

# A highly effective and self-transmissible CRISPR antimicrobial for elimination of target plasmids without antibiotic selection

Panjaporn Wongpayak[1], Orapan Meesungnoen[2], Somchai Saejang[1] and Pakpoom Subsoontorn[1]

[1] Department of Biochemistry, Faculty of Medical Science, Naresuan University, Muang Phitsanulok, Phitsanulok, Thailand
[2] Unaffiliate, Muang Phitsanulok, Phitsanulok, Thailand

## ABSTRACT

The use of CRISPR/Cas (Clustered Regularly Interspaced Short Palindromic Repeats/ CRISPR associated protein) for sequence-specific elimination of bacteria or resistance genes is a powerful tool for combating antibiotic resistance. However, this approach requires efficient delivery of CRISPR/Cas DNA cassette(s) into the targeted bacterial population. Compared to phage transduction, plasmid conjugation can deliver DNA to a broader host range but often suffers from low delivery efficiency. Here, we developed multi-plasmid conjugation systems for efficient CRISPR/Cas delivery, target DNA elimination and plasmid replacement. The CRISPR/Cas system, delivered *via* a broad-host-range R1162 mobilizable plasmid, specifically eliminated the targeted plasmid in recipient cells. A self-transmissible RK2 helper plasmid facilitated the spread of mobilizable CRISPR/Cas. The replacement of the target plasmid with another plasmid from the same compatibility group helped speed up target plasmid elimination especially when the target plasmid was also mobilizable. Together, we showed that up to 100% of target plasmid from the entire recipient population could be replaced even at a low (1:180) donor-to-recipient ratio and in the absence of transconjugant selection. Such an ability to modify genetic content of microbiota efficiently in the absence of selection will be critical for future development of CRISPR antimicrobials as well as genetic tools for *in situ* microbiome engineering.

## INTRODUCTION

Antibiotics have revolutionized modern medicine and saved millions of lives. Unfortunately, frequent uses of antibiotics often disturb beneficial microorganisms and accelerate the rise of antibiotic resistance. In 2019, the US alone had over 2.8 million patients and over 35,000 deaths due to antibiotic resistant bacteria infection (*Centers for Disease Control and Prevention, 2020*). The uses of antimicrobials based on CRISPR (Clustered Regularly Interspaced Short Palindromic Repeats) have been proposed as a strategy for combating antibiotic resistance by target elimination of pathogenic bacteria or

Corresponding author
Pakpoom Subsoontorn,
pakpoomsu@nu.ac.th

curing them of resistance or virulence gene(s) (*Citorik, Mimee & Lu, 2014*; *Bikard et al., 2014*). The CRISPR antimicrobial consists of two components: Cas (CRISPR-associated protein) endonuclease that cleaves DNA target and guide RNA (gRNA) that specifies DNA target sequence for Cas endonuclease. This antimicrobial strategy is highly specific and programmable. Cas endonuclease only eliminates or re-sensitizes bacteria that has DNA target site for gRNA. gRNA specificity can distinguish between bacterial strains that differ by as little as a single base (*Citorik, Mimee & Lu, 2014*). CRISPR antimicrobials are also highly programmable as gRNA can easily be re-designed to target different sequences simply by changing 10–20 bases at a spacer region. Previous works showed that CRISPR antimicrobial can be delivered to selectively kill or re-sensitize pathogenic bacteria such as *Escherichia coli* (*Citorik, Mimee & Lu, 2014*; *Yosef et al., 2015*; *Dong et al., 2019*; *Wang et al., 2019*; *Ruotsalainen et al., 2019*), *Staphylococcus aureus* (*Bikard et al., 2014*), *Salmonella enterica* (*Hamilton et al., 2019*) and *Enterococcus faecalis* (*Rodrigues et al., 2019*).

CRISPR antimicrobial is transferred to target bacteria as a DNA cassette that has a *cas* endonuclease gene and a gRNA gene or a CRISPR locus. Therefore, efficient DNA delivery is critical for effective CRISPR antimicrobials. Transformation, a direct delivery of DNA into cells, is the most straightforward approach. However, only certain bacteria were confirmed to be naturally competent and usually only a small fraction of the population can take up DNA cassette directly (*Johnsborg, Eldholm & Håvarstein, 2007*). Thus, nearly all previous CRISPR antimicrobial works demonstrated the delivery of CRISPR antimicrobials to bacteria populations *via* phage transduction or plasmid conjugation (Table S1) (*Bikard et al., 2014*; *Citorik, Mimee & Lu, 2014*; *Yosef et al., 2015*; *Hullahalli, Rodrigues & Palmer, 2017*; *Dong et al., 2019*; *Wang et al., 2019*; *Hamilton et al., 2019* and *Rodrigues et al., 2019*). Phage is small relative to bacteria and can replicate rapidly to a high phage-to-target bacteria ratio. However, phage has a limited host range and bacteria often quickly develop resistance to phage. On the contrary, many broad-host-range plasmids could conjugatively transfer DNA cassettes to various species of bacteria (*Jain & Srivastava, 2013*). Phage transduction relies on the compatibility between surface proteins on phage particles and on bacterial cells. Thus, bacteria can become resistant to phage simply by altering bacteria surface proteins (*Azam & Tanji, 2019*). On the other hand, conjugative transfer does not involve interaction with specific bacterial surface proteins (*Gruber et al., 2016*). Compared to phage transduction, it is less likely that the resistance to conjugative transfer will emerge. While phages deliver DNA more efficiently to planktonic bacteria in solution, conjugative transfer is better for DNA delivery to a dense bacteria population such as in biofilms on solid surfaces.

Practical uses of conjugative CRISPR antimicrobial delivery are currently limited by DNA transfer efficiency. Nearly all previous studies measured 'efficiency' for CRISPR antimicrobials by comparing the number of surviving transconjugants when a gene inside a recipient was targeted by CRISPR to when there was no CRISPR target (Table S1). Such calculation directly measured efficiency of CRISPR mediated cell death or target plasmid loss but did not account for CRISPR delivery efficiency. In fact, the majority of the targeted cells might not even receive CRISPR antimicrobial cassettes. All studies that
demonstrated CRISPR antimicrobial efficiency without transconjugant selection used at least 1:1 donor-to-recipient ratio and, in some study, up to over 300:1 donor-to-recipient ratio (*Citorik, Mimee & Lu, 2014*). The use of antibiotics to select transconjugant or excessive use of donor cells relative to recipient population is not practical for *in vivo* or *in situ* applications. Some studies proposed the use of self-transmissible conjugative plasmid to deliver CRISPR antimicrobials more efficiently (*Hamilton et al., 2019*; *Ruotsalainen et al., 2019*; *Dong et al., 2019*). Nonetheless, CRISPR antimicrobial efficiency at low donor-to-recipient ratio has not been measured. Moreover, conjugation machineries on the self-transmissible conjugative plasmid may mobilize undesirable resistance or virulence plasmids thereby negating the benefit of CRISPR antimicrobial systems.

Here, we demonstrated highly efficient CRISPR antimicrobial delivery systems based on conjugation and mobilization of multi-plasmids. The systems consisted of an RK2 (*Kolatka et al., 2010*) based self-transmissible helper plasmid and an R1162 (*Meyer et al., 1985*) based mobilizable plasmid encoding CRISPR/Cas cassettes. These two plasmids can be co-transferred to the majority of recipient cells in the absence of antibiotic selection and even at a low (1:180) donor-to-recipient ratio. We explored the consequence of having the CRISPR target plasmid being mobilized by the helper plasmid. We showed that it was possible to also co-deliver an additional plasmid from the same incompatibility group as the target plasmid. The target plasmid in 100% of recipients can be replaced with the new plasmid, thereby providing an additional layer of defense against the spread of the target plasmid. This strategy would enable the applications of conjugatively delivered CRISPR antimicrobials in modulating bacterial populations where transconjugant selection or the use of high donor-to-recipient ratios is impractical.

## MATERIALS & METHODS

### Bacterial strains and plasmids

Donor host cells in all experiments were *E. coli* EcGT2 provided by Dr. Harris H. Wang, University of Columbia, USA (*Ronda et al., 2019*). Recipient host cells in all experiments were *E.coli* SAR08 provided by Dr. Ellen Zechner, University of Graz, Austria (*Reisner, Wolinski & Zechner, 2012*). pgNDM1 is identical to plasmid pMM441 (#61271; Addgene, Watertown, MA, USA) provided by Dr. Timothy K. Lu, MIT, USA (*Citorik, Mimee & Lu, 2014*). pEMPTY was modified from pMM441 by replacing its CRISPR cassette with a gRNA gene whose spacer region was flanked by BsaI restriction sites. pgGFP was constructed by cloning a spacer sequence targeting a green fluorescent protein (GFP) gene into BsaI sites of pEMPTY. ptNDM1 and ptGFP were built by cloning an NDM1 fragment and a GFP gene, respectively, into the backbone of pSEVA231-CRISPR, provided by Dr. Victor de Lorenzo, Nacional de Biotecnología (CSIC) in Madrid, Spain (*Aparicio, de Lorenzo & Martínez-García, 2018*). The NDM1 fragment was amplified from *Acinetobacter baumannii* AB377 provided by Dr. Sutthirat Sitthisak, Naresuan University, Thailand. pHELP is identical to plasmid pRL443 provided by C. Peter Wolk, Michigan State University, USA (*Elhai et al., 1997*). Additional details about plasmid sequences and bacterial genotypes were provided in the Supplementary Materials (Table S2 and Supplementary Files).

## Constructions of donor and recipient cells

pHELP was delivered to donor cells *via* conjugation (see below). pgNDM1, pgGFP, ptNDM1 and ptGFP were introduced to donor cells *via* TSS chemical transformation (*Chung, Niemela & Miller, 1989*). All plasmids were introduced to recipient cells *via* conjugation from *E.coli* EcGT2 donor cells and selected on Luria-Bertani (LB) agar with appropriate antibiotic and without diaminopimelic acid (DAP).

## Media and culture conditions

Unless otherwise noted, *E. coli* were grown in LB broth or agar plates (LB with 15 g agar/L) at 37 °C. Where indicated, antibiotics were added to the selection medium to the following final concentrations: 25 µg/mL of chloramphenicol (Cm) 25 µg/ml of kanamycin (Km) 30 µg/mL of gentamicin (Gm) and 10 µg/mL of tetracycline (Tc). *E. coli* EcGT2 were grown in LB with 50 µM of DAP.

## Conjugation assay

Donors and recipients were each inoculated from a single colony into LB broth with appropriate antibiotics and DAP (for EcGT2 donor cells) and cultured overnight in a 37 °C 200 rpm shaker. Saturated overnight cultures were regrown at 1% dilution ratio in fresh LB broth with appropriate antibiotics and DAP (for EcGT2 donor cells) until $OD_{600}$ reached 0.3–0.7. Cells were spun down at 5,000 g for 5 min and resuspended with 1 mL of LB broth three times in order to wash off residue antibiotics. The volumes of LB broth used in the final resuspension were adjusted so that both donor and recipient cells had $OD_{600}$ ~ 0.3l. Unless otherwise noted, donor cells and recipient cells were mixed at 1:100 ratio by volume. For EcGT2 donor and SAR08 recipient, $OD_{600}$ of 1 is equivalent to 1.14E + 09 and 0.63E + 09 CFU/m, respectively (see Table S4). Thus, donor-to-recipient ratio of 1:100 by volume (at the same $OD_{600}$) is approximately equivalent to donor-to-recipient ratio of 1:180 by cell number. 10 µL of mixed cells were dropped on 1 × 1 cm nitrocellulose membrane on LB agar with DAP. Unless otherwise noted, conjugation between donor and recipient on membranes were allowed to proceed for 18 h at 37 °C. Following conjugation, each membrane was placed in a 1.5 mL tube containing 1 mL of phosphate buffer saline and vortexed for 10 s three times to remove the bacteria from the membrane. The supernatant was serially diluted and plated on LB agar with appropriate antibiotic selection. Agar plates were incubated overnight at 37 °C for 16–24 h. Colonies were counted manually. Conjugation frequencies were measured as the number of transconjugants per recipient. Unless otherwise noted, each experiment was performed in triplicate.

## Data visualization and statistics

Data visualization and statistical analysis were performed using R statistical computing software (https://www.r-project.org/). Scatter plots were generated using ggplot2 package. Student's t-test of the difference between two data sets were conducted using t.test function in R. For multiple comparisons, we performed ANOVA with Tukey's test for *post hoc* analysis using aov and TukeyHSD function in R.

## RESULTS

### Testing conjugatively delivered CRISPR/Cas for target plasmid elimination

A system was set up for conjugative delivery of CRISPR/Cas plasmid (pCRISPR) from donor cells and testing functionality of CRISPR/Cas in recipient cells (Fig. 1A, Fig. S1). pCRISPR was a plasmid encoding a R1162 broad-host-range origin of replication/transfer, a chloramphenicol resistance marker, a *cas9* endonuclease gene, a tracrRNA gene and a CRISPR cassette or a gRNA. pTarget was a plasmid encoding a pBBR1 origin of replication, RK2 origin of transfer, a kanamycin resistance marker and a target gene for CRISPR/Cas. Once in a recipient cell, crRNA, tracrRNA and Cas9 expressed from pCRISPR can together cut the target gene and eliminate pTarget (Fig. 1B).

The spacer sequence of CRISPR cassette or gRNA could be re-designed for different target genes. In the first experiment, we had two versions of pCRISPR, pgNDM1 and pgGFP, with a spacer targeting New Delhi metallo-beta-lactamase 1 (NDM1) gene and Green Fluorescent Protein (GFP) gene, respectively. A total of two versions of pTarget, ptNDM1 and ptGFP, had a fragment of NDM1 and a GFP gene, respectively. We delivered pCRISPR from a donor host cell, EcGT2, which had an RK2 conjugation machinery integrated to the genome. The donor host cell was also auxotrophic for an essential cell-wall component diaminopimelic acid (DAP), thus requiring DAP supplementation in the growth media. This allowed us to select against donors simply by not adding DAP to growth media. Four mating experiments were conducted between donor cells (with pgNDM1 or pgGFP) and recipient cells (with ptNDM1 or ptGFP) (Fig. 1C). We showed that the number of pCRISPR transconjugants that still had pTarget (Fig. 1C, triangle) is less than the total number of transconjugant (Fig. 1C, circle) when the spacer sequence on pCRISPR matched the target gene on pTarget. Therefore, we concluded that CRISPR/Cas system expressed from pCRISPR could cut and eliminate specific pTarget in the recipient cell.

Next, we attempted to show that our mobilizable CRISPR/Cas can eliminate the target gene in a mixed recipient cell population (Fig. 1D). We conducted two mating experiments using EcGT2 donors with pgNDM1 or pgGFP. For both experiments, a recipient population consisted of cells with ptNDM1 or with ptGFP mixed together at 1:1 ratio. The percentage of green (GFP expressing) cells indicated the percentage of cells with ptGFP in the recipient population. Among transconjugants that maintained pTarget, the percentage of green cells increased to near 100 percent when pgNDM1 was delivered but decreased when pgGFP was delivered (Fig. 1E, triangle). This result implies that the delivery of pgNDM1 completely eliminated ptNDM1 in transconjugants. All surviving transconjugants with pTarget were green, *i.e.*, having ptGFP. On the contrary, the delivery of pgGFP eliminated most of ptGFP in transconjugants. In this case, most surviving transconjugants with pTarget were white, *i.e.*, having ptNDM1. Therefore, mobilizable CRISPR/Cas can specifically knockdown subpopulation, thereby altering the composition of the mixed population.

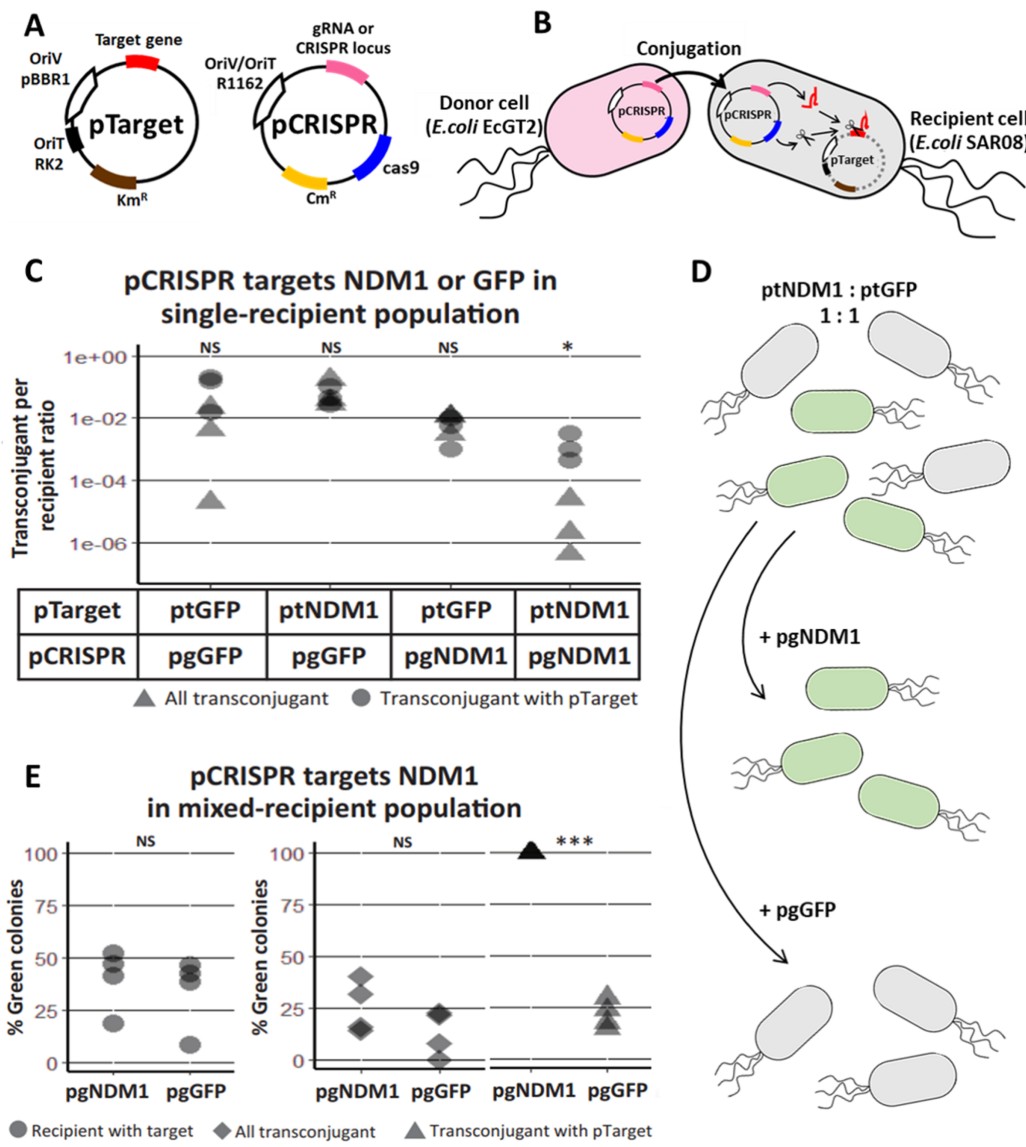

**Figure 1 Mobilizable CRISPR/cas eliminates specific plasmid targets in bacteria population.**
(A) A schematic of a CRISPR/Cas mobilizable plasmid (pCRISPR) and a target plasmid (pTarget). CRISPR plasmid has a *cas9* endonuclease gene, CRISPR locus with a spacer or a guide RNA (gRNA or CRISPR locus) gene for targeting a specific targeted gene, R1162 vegetative plasmid origin and origin of transfer (OriV/OriT-R1162) and a chloramphenicol resistance gene ($Cm^R$). Target plasmid has a target gene, pBBR1 vegetative plasmid origin (OriV-pBBR1), RK2 origin of transfer (OriT) and a kanamycin resistance gene ($Km^R$). (B) A schematic of pCRISPR mobilized to a recipient cell and eliminating a target plasmid. pCRISPR was mobilized to the recipient cell by a conjugation machinery encoded in the genome of the donor cell (EcGT2). Once in the recipient cell, pCRISPR expressed Cas9 endonuclease and crRNA or gRNA that cut a target DNA on the pTarget. DNA cut led to degradation of the pTarget. (C) Conjugative delivery of pCRISPR targeting a New Delhi metallo-beta-lactamase one gene (pgNDM1) and pCRISPR targeting a green fluorescent protein gene (pgGFP) caused specific clearance of the pTarget with NDM1 fragment (ptNDM1) and target plasmid with GFP gene (ptGFP), respectively. *E. coli* EcGT2 donor cells containing pgNDM1 or pgGFP were mated at a donor-to-recipient ratio of 1:18 for 18 h with *E. coli* SAR08 recipient cells that contain ptNDM1 or ptGFP. Cultures were plated on Luria–Bertani (LB) with chloramphenicol (Cm) to select for all transconjugant cells, LB with kanamycin (Km) to select for recipient cell with pTarget and LB containing both Cm and Km to select for transconjugant cells that still contained pTarget. Points represent independent experimental replicates. Circular points represent the

**Figure 1** (continued)
ratio of all transconjugant cells per recipient cells. Triangular points represent the ratio of transconjugant cells that still contained a pTarget per recipient cells. *P*-value from ANOVA with Tukey's test for *post hoc* analysis of the difference between the average values of circular points and of triangular points: ns ≥ 0.05, * < 0.05, ** < 0.01, *** < 0.001. (D) A schematic of pCRISPR mobilized to a recipient cell and eliminating the target plasmid in a mixed population. (E) CRISPR/Cas could eliminate target plasmid that have specific DNA sequences without disrupting other bacteria. Donors (EcGT2) containing pgNDM1 or pgGFP were mated to mixed recipients with ptNDM1 or ptGFP at a donor-to-recipient ratio of 1:18 for 18 h. Mated cells were plated on agar media with Cm, with Km or with Cm and Km to quantify the number of total recipients and of transconjugants that received pCRISPR, respectively. Each point in the figure represents pCRISPR transfer frequency calculated from each independent experiment. *P*-value from Student's t-test for the difference between the average % green colonies of experiment using pgNDM1 and experiment using pgGFP: ns ≥ 0.05, *<0.05, ** < 0.01, *** < 0.001.

Among all transconjugants, the percentages of green cells in both experiments were only slightly different (Fig. 1E, rhombus). This could be explained by the fact that CRISPR/Cas mediated elimination of pTarget did not lead to cell death. The delivery of pgNDM1 may eliminate ptNDM1 in a recipient cell but the transconjugant still survived and remained as part of white cell population. The delivery of pgGFP led to elimination of ptGFP. Even if the cell survives, GFP signal is lost, resulting in reduction of green colonies (Fig. 1E triangle and Fig. S2). Without transconjugant selection, the percentages of green cells in both experiments were similar (Fig. 1E, circle). This is because most recipient cells did not receive pCRISPR and thus were not affected by CRISPR/Cas (Table S3). These results imply that the efficiency of mobilizable CRISPR/Cas in altering cell populations was limited by efficiency of DNA delivery.

## Self-transmissible CRISPR/cas delivery systems

Since conjugative transfer of pCRISPR relied on conjugation machinery in the donor (EcGT2) genome, pCRISPR can only be mobilized to recipients immediately adjacent to donors. In order to deliver pCRISPR further, we introduced pHELP, a self-transmissible plasmid that can also mobilize pCRISPR from a transconjugant to the next recipients (Fig. 2A). pHELP is an RK2 conjugative plasmid with a tetracycline resistance marker, RK2 origin of transfer and conjugation machinery that can mobilize both RK2 and R1162 origin of transfer. By having both pHELP and pCRISPR in donor cells, we expected that pCRISPR can spread to a larger proportion of recipients even in the absence of transconjugant selection.

We conducted mating experiments using EcGT2 donors with pCRISPR alone or with both pCRISPR and pHELP (Fig. 2B). Note that for all following experiments, we used pgNDM1 as our only pCRISPR. Thus, we referred to this plasmid simply as pCRISPR (and its targeted plasmid, ptNDM1, as pTarget). Donors and recipients were mixed together at 1:18 or 1:180 donor-to-recipient ratio. At both donor-to-recipient ratios, mating experiment with pHELP yielded significantly higher pCRISPR transconjugant per recipient than mating experiment without pHELP (Fig. 2B, rhombus *vs.* triangle). For mating experiment with pHELP, pCRISPR transconjugant per recipient did not significantly decreased as we increased donor-to-recipient ratio from 1:18 to 1:180.

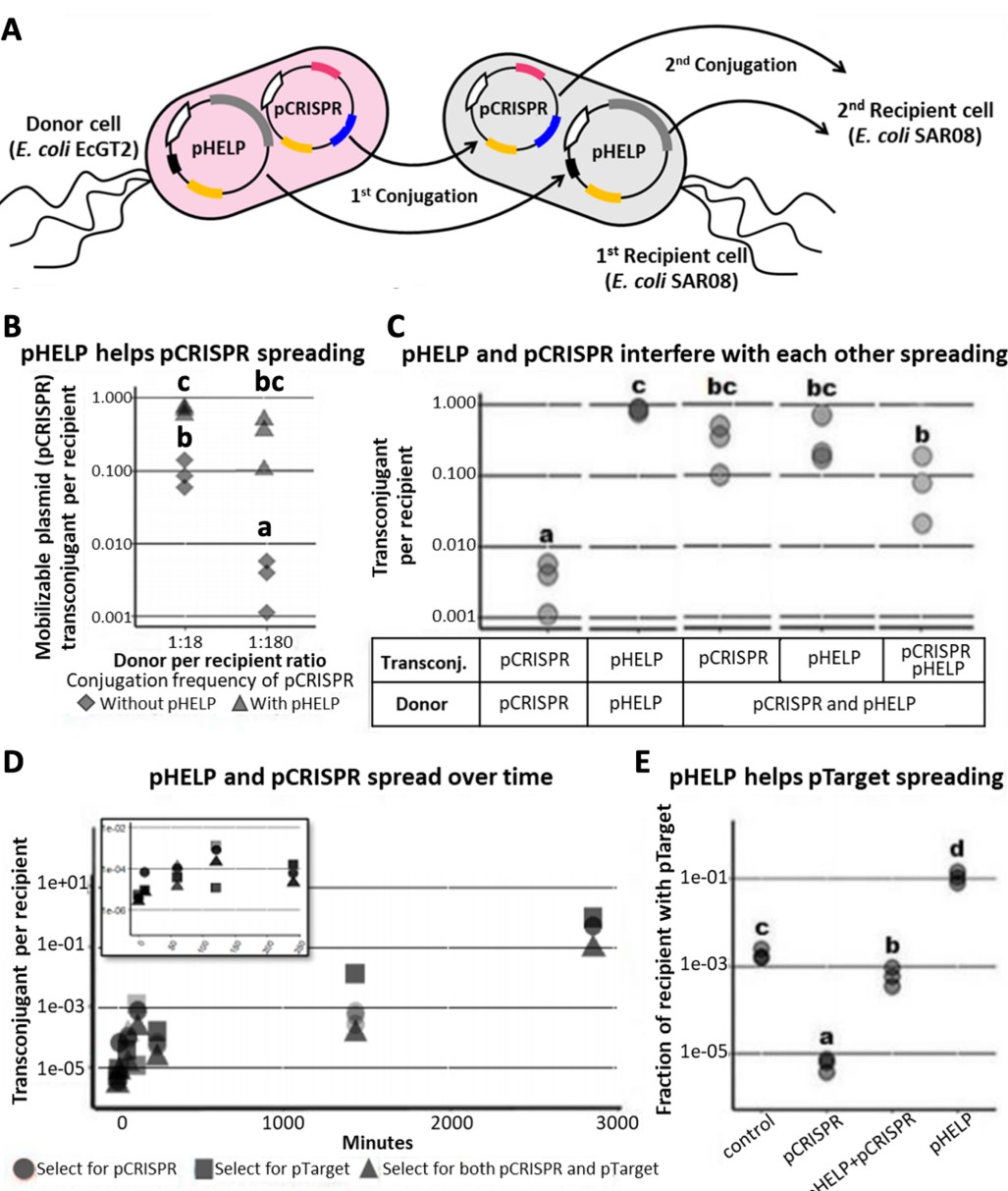

**Figure 2 Self-transmissible RK2 plasmid efficiently delivers DNA to bacteria population even in the absence of transconjugant selection.** (A) A schematic of a multi-plasmid system comprising CRISPR mobilizable plasmid (pCRISPR) and conjugative helper plasmid (pHELP). pCRISPR was mobilized to a recipient cell by conjugation machinery encoded in conjugative plasmid or encoded in the genome of the donor cell (EcGT2). Once in the recipient cell, pHELP can transfer itself and pCRISPR further to secondary recipient cells. (B) Donors (EcGT2) containing conjugative plasmid and mobilizable plasmid (pHELP and pCRISPR) or mobilizable plasmid alone (only pCRISPR) were mated to recipients (SAR08) at a donor-to-recipient ratio of 1:18 or 1:180 for 18 h. Mated cells were plated on agar media without or with Cm to quantify the number of total recipients and the number of transconjugants with pCRISPR, respectively. Each point in the figure represents pCRISPR transfer frequency calculated from each independent experiment. Red and blue points indicate conjugation frequency of pCRISPR without or with pHELP, respectively. Different letters (a, b, c, …) indicate statistically significant differences ($P < 0.05$) between groups according to ANOVA with Tukey's test for *post hoc* analysis. (C) Conjugation frequency of pCRISPR and pHELP transferred alone or in the presence of each other. Each point represents conjugation frequency from an individual replicate experiment. Different letters (a, b, c, …)

**Figure 2** (continued)
above each column indicate statistically significant differences ($P < 0.05$) between groups according to ANOVA with Tukey's test for *post hoc* analysis. (D) The number of transconjugants per recipient at 0, 10, 60, 120, 240, 1,440 and 2,880 min from the beginning of conjugation experiment. Each point represents independent experimental replicate. (E) pHELP facilitated pTarget spread in the population. Donors were mated with recipients at 1:180 ratio. At the beginning of the experiment, 1/100 recipients had pTarget. The ratio between recipients with pTarget and total recipients were measured after 18 h. of mating. Different letters (a, b, c, …) above each column indicate statistically significant differences ($P < 0.05$) between groups according to ANOVA with Tukey's test for *post hoc* analysis.

Therefore, the use of pHELP could allow us to deliver pCRISPR widely in the population even at low donor-to-recipient ratio and in the absence of transconjugant selection.

We performed further experiments to explore how pHELP and pCRISPR affect transfer efficiencies of each other (Fig. 2C). The mating experiments were conducted at 1:180 donor-to-recipient ratio for 18 h on LB agar. We found that pHELP by itself can spread from donors to nearly 100% of recipients. pCRISPR by itself was only delivered, using conjugation machinery in the genome of donor cells, to only 0.3% of all recipients on average. In the presence of pHELP, the pCRISPR transconjugants increased to 31.9% of all recipients on average. On the contrary, the presence of pCRISPR decreased pHELP transconjugants down to 36% on average. In other words, pHELP boosted pCRISPR spreading while pCRISPR hindered pHELP spreading. About 9% of recipients had both pCRISPR and pHELP. To explore the dynamics of pCRISPR and pHELP transfer, we measured the number of transconjugants with pCRISPR, pHELP or both per recipient over time. The experiment started at 1:180 donor-to-recipient ratio; the donor had both pCRISPR and pHELP (Fig. 2D). Early on, pCRISPR appeared to spread faster but were later caught up and surpassed by pHELP. We found that after 2 days (2880 min) over 50% of recipient cells received pCRISPR. Thus, since the percentage of recipients with pCRISPR continued to increase over time, it should be possible to deliver mobilizable CRISPR/Cas plasmid to the entire bacteria population without transconjugant selection.

However, a self-transmissible CRISPR delivery system could also mobilize undesirable plasmids (*e.g.*, antibiotic resistance or virulence plasmids) in the population. In the following experiment, we started with a recipient population in which only 1% of recipient cells had pTarget (representing an undesirable antibiotic resistance plasmid) which had a target site for pCRISPR but can also be mobilized by pHELP. We mated this recipient population to donor cells with only pCRISPR (no self-transmissible helper plasmid), to donor cells with only pHELP or to donor cells with both pHELP and pCRISPR. For the experiment with pCRISPR alone, the percentage of recipient cells with pTarget decreased by over a hundred folds relative to the control experiment (no plasmid in donors). On the contrary, when the donor cells had only pHELP, the percentage of recipient cells with pTarget increased by nearly a hundred folds relative to the control experiment. When both pCRISPR and pHELP were present, the percentage of recipient cells with pTarget remained almost equal to the control experiment. In other words, while pHELP increased overall delivering efficiency of CRISPR/Cas cassette (on pCRISPR in this

case), the target plasmid was eliminated less efficiently. Our simulation results also confirmed the possibility of such scenarios in which the presence of a self-transmissible CRISPR delivery system could hinder rather than help the elimination of the target plasmid (Figs. S5, S6).

## Elimination and replacement of the target plasmid with an incompatible plasmid

To fully utilize a self-transmissible CRISPR antimicrobial system, one has to prevent undesirable spread of the target plasmid by the helper plasmid (pHELP). We reasoned that the existence of a plasmid in a population could slow down or stop the entry of another plasmid from the same incompatibility group. While plasmid incompatibility does not directly blocking plasmid transfer, an incoming plasmid would be unlikely to establish in the population in the absence of selective advantage. In all following experiments, we used ptGFP as "pSubst", a plasmid for replacing ptNDM1 (from now referred to as "pTarget"). pSubst and pTarget belong to the same incompatibility group and should not co-exist stably in the same host cell. We used pgNDM1 as "pCRISPR" which can express CRISPR/Cas specifically for cleaving pTarget but not pSubst.

We demonstrated a multi-plasmid system that can both eliminate an undesirable plasmid (pTarget) and replace it with another plasmid (pSubst) from the same incompatibility group. EcGT2 donors with both pCRISPR and pSubst were mated to recipients that had pTarget. pSubst by itself is unlikely to be able to enter, compete and replace pTarget that has already established in recipient population. However, when the donor has both pCRISPR and pSubst, pCRISPR can first be mobilized into recipients and mediate the degradation of pTarget. Once pTarget is eliminated, pSubst can then be mobilized into the recipient. We measured the fractions of recipients in each stage of plasmid replacement (Fig. 3A). We found that over 96% of recipients did not receive any plasmid (Fig. 3A, cell type X). Among transconjugants (cell type Y, Z and W), the majority of cells had both pCRISPR and pSubst (cell type W). Only a tiny fraction of transconjugants had both pCRISPR and pTarget (cell type Y). Overall, although recipients were unlikely to receive any plasmid, most recipients that received pCRISPR also received pSubst.

Next, we attempted to confirmed that a new plasmid cannot establish in the population that is already saturated with another plasmid from the same incompatibility group. We conducted two mating experiments using (1) EcGT2 donors with pCRISPR and pSubst or (2) EcGT2 donors with pEMPTY and pSubst. pEMPTY was similar to pCRISPR but without a spacer for targeting pTarget. All recipients in both experiments already had pTarget before mating. Donors were mated with recipients at 1:180 donor-to-recipient ratio. We confirmed that transfer efficiency of pEMPTY and pCRISPR were not significantly different (Fig. 3B). When the donors had pCRISPR, we found that 75% of transconjugants received pSubst and turned green (Fig. 3C, rhombus). With pEMPTY in donors instead of pCRISPR, all transconjugants remained white. This experiment confirmed that the replacement plasmid by itself could not enter, compete and replace the target plasmid that had already established in the population. The use of CRISPR plasmid
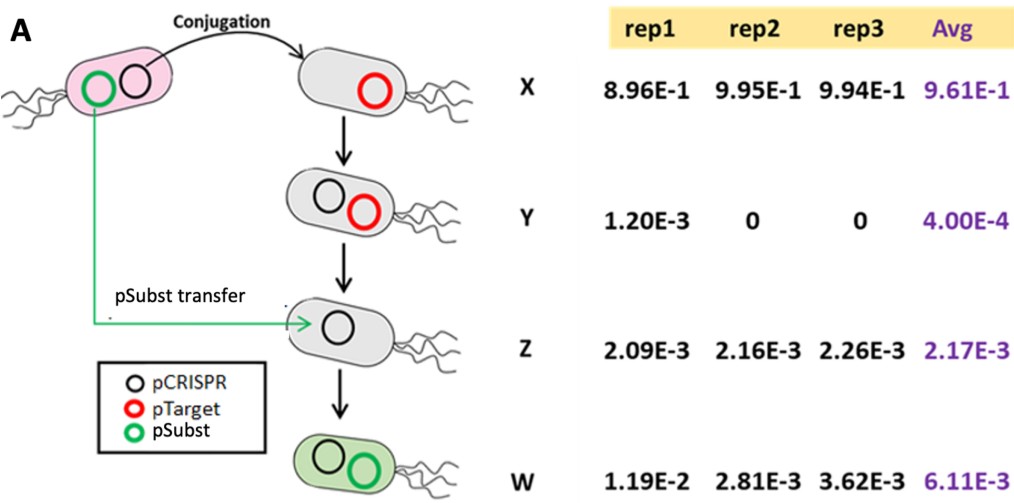

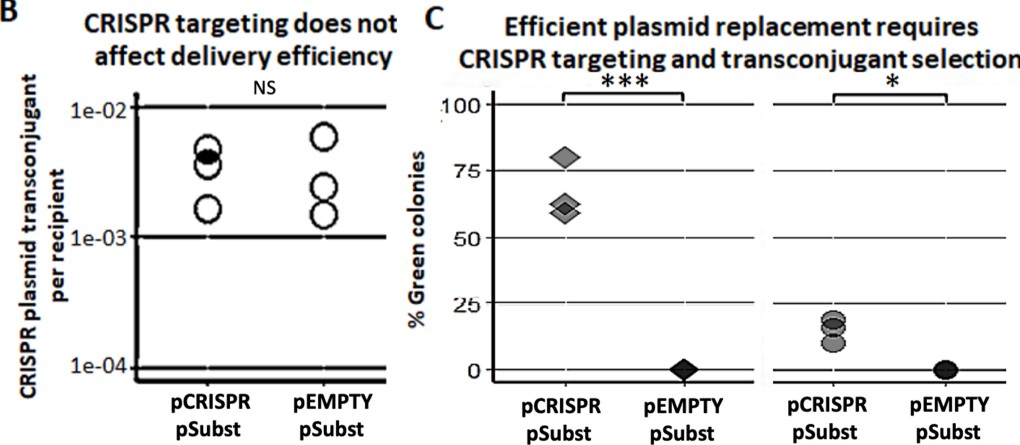

**Figure 3 Mobilizable CRISPR/cas mediates replacement of plasmids from the same incompatibility group.** (A) Mechanism and efficiency of plasmid replacement by mobilizable CRISPR/Cas. (Left) A schematic shows each stage of plasmid transfer, elimination and replacement (X = original recipient cell, Y = recipient cell after receiving pCRISPR, Z = recipient cell after losing pTarget, W = recipient cell after receiving a replacement plasmid (pSubst)). (Right) Each row shows the ratio of recipient cells in each stage of replacement. Each column shows result from each experimental repeat and average values. (B) Donors (EcGT2) containing pCRISPR or pEMPTY and pSubst (GFP expressing) were mated to target recipients containing pTarget at a donor-to-recipient ratio of 1:180 for 18 h. Each point indicates pCRISPR or pEMPTY transfer frequency calculated from each independent experiment. *P*-value from Student's t-test for difference between average transfer frequency of each group: ns ≥ 0.05, * < 0.05, ** < 0.01, *** < 0.001. (C) From the same experiment, the fraction of recipient whose pTarget was replaced by pSubst was measured as the fraction of colonies that express GFP after mating. *P*-value from Student's t-test for the difference between the average % green colonies of experiment using pCRISPR and experiment using pEMPTY: ns ≥ 0.05, * < 0.05, ** < 0.01, *** < 0.001.

to eliminate pTarget in recipients was critical for the replacement. Moreover, the majority of recipients that received pCRISPR also received pSubst. In other words, these two plasmids were likely to be transferred together. Nonetheless, like in the previous experiment (Fig. 3A), only a small fraction of recipients received any plasmid from the

donors in the absence of transconjugant selection. For this reason, only a small percentage of all recipients received pSubst and turned green (Fig. 3C, circle).

To demonstrate generalizability of a plasmid substitution system, we performed an experiment similar to that shown in Fig. 3 but using a different pair of target and substitution plasmids (Fig. S8). These new pairs of plasmids have RK2 origin of replication (rather than pBBR1) and different backbone sequences from pTarget and pSubst. We confirmed that functional CRISPR/Cas was still critical for elimination and replacement of the target plasmid. Additionally, we performed conjugation experiments of EcGT2 donors with pTarget and pEMPTY to recipients without or with pSubst. Our results indicated that the presence of pSubst could prevent the delivery of pTarget to recipients albeit incompletely (Fig. S9).

## Self-transmissible CRISPR/cas delivery and target plasmid replacement

The utilization of CRISPR antimicrobial requires high DNA transfer frequency. Thus far, we have shown that the use of self-transmissible helper plasmid (pHELP) can greatly increase transfer efficiency of a CRISPR plasmid (*e.g.*, pCRISPR) even at low donor-to-recipient ratio and without antibiotic selection (Figs. 2B and 2C). Nonetheless, pHELP could also promote the dissemination of undesirable mobilizable plasmid we would like to eliminate (Fig. 2E). We have also shown that the use of CRISPR plasmid can facilitate the replacement of the target plasmid with a replacement plasmid from the same incompatibility group (Fig. 3). Here, we would like to explore whether self-transmissible CRISPR delivery system together with the use of plasmid incompatibility could drive complete elimination of undesirable mobilizable plasmid.

Our experimental system consisted of the following plasmids and *E. coli* strains (Fig. 4A). pTarget represented, an undesirable target plasmid (*e.g.*, resistance or virulence plasmid) we would like to eliminate from the bacteria population. pSubst represented a replacement plasmid from the same incompatibility group as pTarget (but without any harmful genetic element). pCRISPR represented a CRISPR plasmid targeting pTarget. pHELP was a self-transmissible helper plasmid that can mobilize pCRISPR, pTarget and pSubst. An *E. coli* cell in the population can be classified into different "types" according to the type(s) of plasmid(s) the cell possessed. For example, a cell type A has no plasmid, a cell type B has a pTarget and a cell type G has both pTarget and pHELP. Any *E. coli* strain in the population can receive pHELP, pCRISPR, pTarget and pSubst thereby converting into another strain (blue, black, red and green arrow, respectively in Fig. 4A). For example, an *E. coli* strain-E that receives pCRISPR becomes an *E. coli* strain-J. pTarget and pSubst belong to the same incompatibility group. We previously showed that pSubst cannot enter and establish in the recipient that already had pTarget. Similarly, it is likely pSubst could also hinder the establishment of pTarget. Additionally, if pCRISPR and pTarget are in the same host cell, pTarget will be eliminated (dashed red arrow in Fig. 4A). If all four plasmids are present in the population, all recipients in the population are expected to end up receiving three plasmids (*e.g.*, pCRISPR, pSubst and pHELP) while pTarget are eliminated completely from the population. We conducted two mating

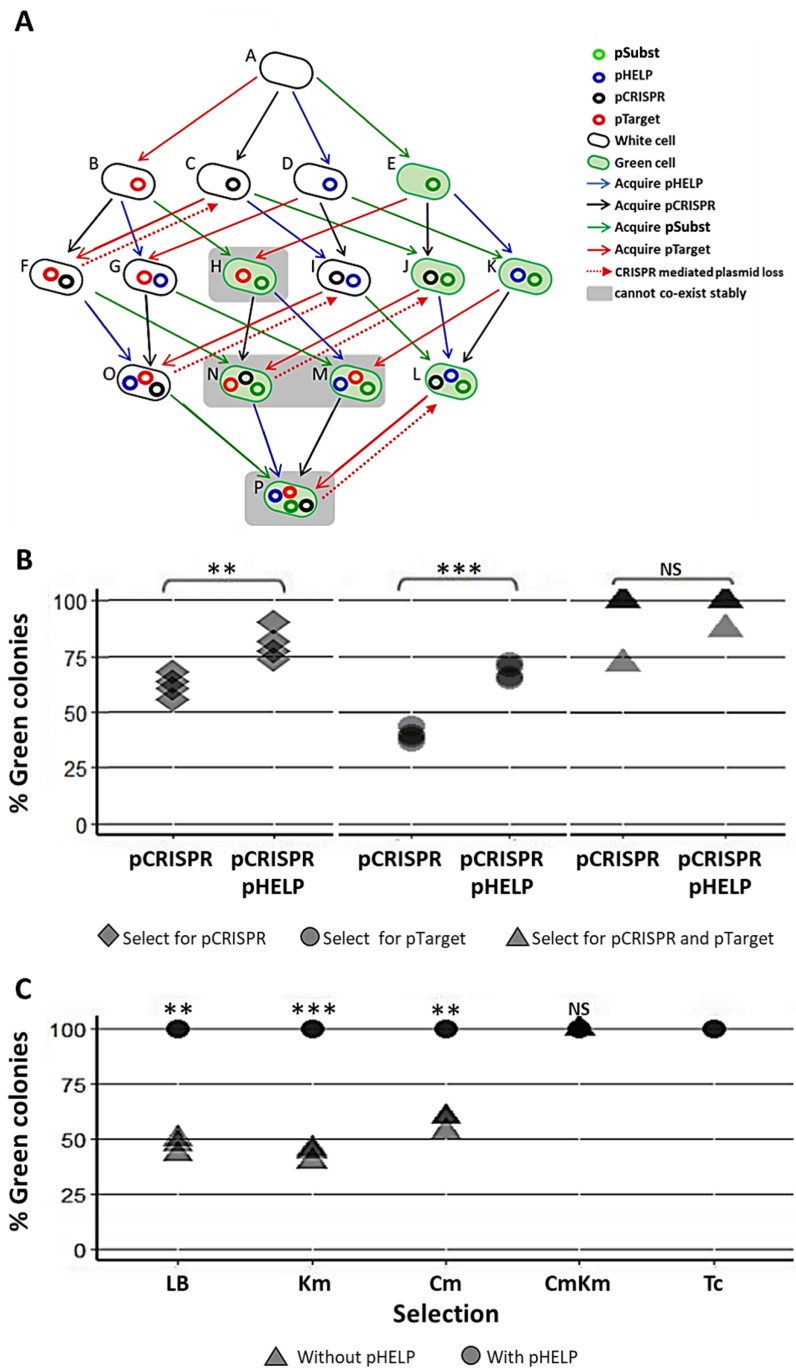

**Figure 4 Replacing plasmid in the entire population *via* self-transmissible CRISPR/cas and incompatibility plasmid system.** (A) A schematic of all possible plasmid acquisitions and losses in bacterial population harboring pCRISPR, pTarget, pSubst and pHELP. Arrows represent transition between different cell "types"; each cell type has a different combination of plasmids. (B) Donors (EcGT2) containing pCRISPR and pHELP were mated to mixed recipients containing pTarget and pSubst at a donor-to-recipient ratio of 1:180 for 18 h. At the beginning of conjugation, 50% of recipient population had pSubst while the rest had pTarget. After the experiment, mated cells were plated on LB agar with Cm (select for pCRISPR), Km (select for pTarget or pSubst) or Cm and Km (select for both pCRISPR and pTarget or pSubst). The percentages of green colonies indicate the percentages of selected recipient

**Figure 4** (continued)
harboring pSubst. Each point shows percentage of green colonies from each independent experiment. (C) Donors (EcGT2) containing pCRISPR and pSubst, with or without pHELP, were mated to recipients containing pTarget at a donor-to-recipient ratio of 1:180 for 4 days. Mated cells were plated on LB agar without antibiotic (no selection), with Km (select for pSubst or pTarget), Cm (select for pCRISPR), both Cm and Km, or with Tc (select for pHELP). Circular and triangular points showed the results from mating experiment using the donor without or with pHELP, respectively. Each point represents the percentages of green colonies calculated from each independent experiment. *P*-value from ANOVA with Tukey's test for *post hoc* analysis of the difference between the average % green colonies from experiment without and with pHELP: ns ≥ 0.05, * < 0.05, ** < 0.01, *** < 0.001.

experiments using (1) EcGT2 donors with only pCRISPR or (2) EcGT2 donors with pHELP and pCRISPR. For both experiments, the recipient population consisted of cells with pTarget and cells with pSubst mixed at 1:1 ratio.

The EcGT2 donors were mated with mixed recipients at 1:180 donor-to-recipient ratio. We experimented on three different selection schemes: (1) selecting for pCRISPR alone, (2) selecting for pTarget or pSubst, (3) selecting for both pCRISPR and pTarget or pSubst. Unless selecting for both pCRISPR and pSubst, mating experiment in the presence of pHELP led to significantly higher percentage of cells that had pSubst. Notably, without selecting for pCRISPR, the percentage of cells with pSubst remained near 50% unless pHELP was present. Hence, pHELP helped mobilize pCRISPR to eliminate pTarget in recipients as well as mobilizing pSubst to replace pTarget (Fig. 4B).

Next, we conducted two mating experiments using (1) EcGT2 donors with pCRISPR and pSubst or (2) EcGT2 donors with pHELP, pCRISPR and pSubst. For both experiments, the EcGT2 donors were mated at 1:180 donor-to-recipient ratio to recipients that has pTarget. We experimented on five different selection schemes: (1) non selection, (2) selecting for pTarget or pSubst, (3) selecting for pCRISPR, (4) selecting for both pCRISPR and pTarget or pSubst and (5) selecting for pHELP. We found that, for the donor with pHELP, 100% of recipients had pTarget replaced by pSubst even in the absence of antibiotic selection after 4 days of conjugation (Fig. 4C). We confirmed that all colonies that received pSubst had eventually lose pTarget (Fig. S4). In computer simulation, we also showed that the presence of both pHELP, pCRISPP and pSubst could all contribute to complete elimination of pTarget (Fig. S7).

# DISCUSSION

Our work addressed two fundamental challenges in the application of conjugation-based CRISPR antimicrobials. First, compared to phage-based delivery, conjugative delivery efficiency is often low. Thus, previous demonstrations of this delivery approach used a high donor-to-recipient ratio together with antibiotic selection of transconjugant in order to highlight CRISPR antimicrobial effects (Table S1) (*Bikard et al., 2014*; *Citorik, Mimee & Lu, 2014*; *Yosef et al., 2015*; *Hullahalli, Rodrigues & Palmer, 2017*; *Dong et al., 2019*; *Wang et al., 2019*; *Hamilton et al., 2019* and *Rodrigues et al., 2019*). Here, we solved this problem by using a self-transmissible helper plasmid that enabled secondary transmission of CRISPR antimicrobial from transconjugant to other recipient cells. However, the

self-transmissible helper plasmid system led to the second problem: the spread of undesirable plasmid, *e.g.*, plasmid containing virulence or antibiotic resistance gene(s). To solve this problem, we co-delivered to the recipient population a replacement plasmid ("pSubst ") from the same incompatibility group as the undesirable plasmid ("pTarget"). The presence of took up available recipient cells thereby hindering the spread of the target plasmid in the population. Our final system combined the delivery of self-transmissible helper plasmid and the replacement plasmid. We demonstrated that our system could eliminate and replace a target undesirable plasmid in 100% of the recipient population without antibiotic selection and using a low (1:180) donor-to-recipient ratio.

The use of self-transmissible plasmid to increase CRISPR antimicrobial delivery efficiency had previously been demonstrated (*Hamilton et al., 2019*; *Dong et al., 2019*; *Ruotsalainen et al., 2019*). However, the key differences between our study and these previous studies are as followings: First, *Hamilton et al. (2019)*, *Dong et al. (2019)* had a CRISPR/Cas cassette on a self-transmissible plasmid while our system has a CRISPR/Cas cassette on a mobilizable plasmid co-transferred with a self-transmissible helper plasmid. The separation between CRISPR/Cas cassette and conjugative machinery on two different plasmids offers more modularity in construction process and plasmid copy number control. Nonetheless, such separation comes with a challenge that conjugative machinery and CRISPR/Cas cassette might not always be transferred together in recipient population. The delivery system by *Ruotsalainen et al. (2019)* had CRISPR/Cas cassette on a mobilizable plasmid separated from another self-transmissible helper plasmid. The authors reported that the mobilizable CRISPR/Cas plasmid failed to spread to the most recipients. On the contrary, when given enough time, our system could deliver CRISPR/Cas cassette to the majority of recipients. Second, these previous works did not provide evidence that the use of self-transmissible CRISPR plasmid(s) can increase CRISPR antimicrobial efficiency. Specifically, *Hamilton et al. (2019)* only showed that self-transmissible CRISPR plasmid can be transferred to recipients more effectively than mobilizable CRISPR plasmid. However, *Hamilton et al. (2019)* did not show that the percentage of recipients had their target gene eliminated by self-transmissible CRISPR plasmids more efficiently, comparing to target elimination by mobilizable CRISPR plasmids. *Ruotsalainen et al. (2019)* did not measure the extent to which CRISPR antimicrobial efficiency of the system with the self-transmissible helper plasmid was higher than the efficiency of the system without. *Dong et al. (2019)* used a high donor-to-recipient ratio (1:1) but could only achieve 1.6-fold increase in a target cell elimination efficiency when using self-transmissible CRISPR plasmid. On the other hand, we showed that we can double CRISPR mediated plasmid elimination and replacement in recipient population from 50% to 100%, despite using low donor-to-recipient ratio (1:180).

The efficiency of RK2-based self-transmissible plasmid measured in our study is similar to previously reported values (*Eden & Blakemore, 1991*; *Cabezon, Lanka & De la Cruz, 1994*; *Baharoglu, Bikard & Mazel, 2010*; *Gama, Zilhão & Dionisio, 2017*). *Baharoglu, Bikard & Mazel (2010)* measured conjugation efficiency of RK2 transferred between *E. coli* on solid agar using 1:1 donor-to-recipient ratio. Over the course of 24 h, RK2-based plasmid can be transferred to 100% of recipients in the absence of antibiotic selection.

In our work, using 1:180 donor-to-recipient ratio, we can deliver RK2-based plasmid to 100% of recipients albeit requiring a longer conjugation time (48 h). In the shorter time scale (1 h), *Gama, Zilhão & Dionisio (2017)* measured the conjugation frequency (transconjugant/sqrt(D * R)) of RK2 transfer between *E. coli* in liquid media using 1:1 donor-to-recipient ratio (*Baharoglu, Bikard & Mazel, 2010*). The conjugation frequency was reported at 1E − 4. Using the same conjugation frequency measure, we estimated our solid phase conjugation frequency (from Fig. 2C) at 3.5E − 4. Therefore, our conjugation frequency was in the same order of magnitude as previously reported. Our measured conjugation frequency was slightly higher than the measured value by *Gama, Zilhão & Dionisio (2017)* possibly because solid phase conjugation tends to be more efficient than liquid phase conjugation.

All CRISPR antimicrobial studies so far relied on mobile genetic elements, phages or conjugative plasmids, to deliver CRISPR cassettes to bacterial populations (Table S1). The use of self-transmissible mobile genetic elements could boost CRISPR delivery efficiency but also carries the risk of spreading undesirable genes such as antibiotic resistance or virulence genes in the population. However, these studies had target genes on genomes or non-mobilizable plasmids. Thus, the effects of target genes spreading in the context of CRISPR antimicrobial applications have not been explored. In our study, CRISPR targets were encoded on a mobilizable plasmid. While a self-transmissible helper plasmid increased the spread of CRISPR antimicrobial, a CRISPR target was also transferred. The risk of CRISPR target spreading could outweigh the benefit of efficient CRISPR antimicrobial delivery. In our experiment and simulation, we found that the presence of a self-transmissible helper plasmid could lead to the decreasing efficiency of target elimination (Fig. 2E). Specifically, after conjugation, the percentage of recipients with target plasmid in the presence of self-transmissible helper plasmid is approximately 100-fold higher than in the absence of a self-transmissible helper plasmid. Our discovery highlighted a critical issue regarding the undesirable spread of the target genes by the delivery system of CRISPR antimicrobial. The prevalence, magnitude and parameters related to this phenomenon shall be investigated in future research.

In order to prevent the spread of the target plasmid, we attempted to replace it with a replacement plasmid from the same incompatibility group. By co-delivering the CRISPR plasmid and the self-transmissible helper plasmid along with the replacement plasmid, we successfully replaced the target plasmid in 100% of recipients without using antibiotic selection in all experimental triplicate. Previous work reported close to 100% delivering efficiency of an RK2-based plasmid without selection (*Baharoglu, Bikard & Mazel, 2010*). However, it was unclear whether multiple mobilizable plasmids can be co-transferred at this level of efficiency. Cabezon et al measured co-transfer rates of an RSF1010 based mobilizable plasmid and an RK2-based conjugative plasmid (*Cabezon, Lanka & De la Cruz, 1994*). The authors found that 70% of an RK2-based plasmid recipients also received an RSF1010 plasmid. This level of co-transfer efficiency is within the same order of magnitude as our measured RK2/R1162 (pHELP/pCRISPR) co-transfer efficiency over similar conjugation time. Another study by *Ruotsalainen et al. (2019)* measured co-transfer rates of RK2 based conjugative plasmid and p15A based plasmid with RK2 origin of

transfer. Nearly all recipients that received RK2 plasmid did not receive p15A plasmid. Our replacement plasmid (ptGFP) had the same origin of transfer but higher copy origin of replication (*i.e.*, pBBR1). This difference could explain why we observed much higher co-transfer efficiency between a conjugation helper plasmid (pHELP) and a mobilizable replacement plasmid.

In nature, bacteria commonly harbor multiple plasmids (*Dionisio, Zilhão & Gama, 2019*). Interactions among plasmids affect their transfer frequencies and overall fitness in the population. For example, a mobilizable plasmid can only be transferred with help from a conjugative plasmid. On the other hand, the mobilizable plasmid may compete with conjugative plasmid for conjugation machinery thereby reducing the transfer rate of the conjugative plasmid. Previous works also reported that the presence of a conjugative plasmid in the donor or recipient cells may inhibit or facilitate the transfer of another conjugative plasmid (*Gama, Zilhão & Dionisio, 2017*). However, it was still unclear how the presence of mobilizable plasmid(s) may help or hinder the spread of other mobilizable plasmid(s). Our work demonstrated a possible scenario in which the presence of one mobilizable plasmid is critical for the spread of another mobilizable plasmid. Specifically, CRISPR/Cas expressed by the first mobilizable plasmid eliminated target plasmid in recipients. Then, the second mobilizable plasmid, from the same incompatibility group as the target plasmid, can follow. Without the first mobilizable plasmid, the second mobilizable plasmid can barely establish in the recipient population due to plasmid incompatibility. In other words, CRISPR/Cas systems can be used as a weapon for invading into the population in addition to their well-known function in defending against invading plasmid or other mobile genetics elements. Given that CRISPR/Cas have been found on natural plasmids and that over 25% of known plasmids are mobilizable, we expected that CRISPR/Cas could also play an important role in plasmid cooperation and invasion in nature (*Alderliesten et al., 2020*).

While we have demonstrated efficient CRISPR antimicrobial delivery *via* conjugation, the following limitations need to be addressed in future. First, we have yet to demonstrate efficient CRISPR antimicrobial delivery in non-*E. coli* and in heterogenous multispecies microbiota. Previous meta-analysis showed that decreasing taxonomic relatedness between donors and recipients led to lower conjugation frequency in liquid media but had no significant effect on conjugation frequency on solid media (*Alderliesten et al., 2020*). Therefore, we expected that our strategy for efficient conjugative delivery could still be applicable in non-*E. coli* in biofilms. Second, CRISPR antimicrobial cassettes in the recipient population may mutate and lose functions. The malfunctioning CRISPR antimicrobial could interfere with the delivery of functional CRISPR antimicrobials. For example, the presence of malfunctioning CRISPR plasmid in recipients could prevent the entry of functioning CRISPR plasmids. Future study should explore the approaches for vector self-clearance (*Lazdins et al., 2020*). Third, the use of multi-plasmid systems could lead to unexpected plasmid recombination. Future plasmid design should aim to reduce probability of homologous recombination, for example, by minimizing the homology regions within and between plasmids (*Dionisio, Zilhão & Gama, 2019*). Forth, given that several types of plasmids could exist in a microbial community, targeting just

one type of plasmid may fail to eliminate resistance genes. One also needs to know beforehand exactly the type of plasmid to target. Thus, future work should include an integration of technologies for profiling plasmid types in targeted microbiota (*Dib et al., 2015*) as well as the development of multiplex CRISPR/Cas and substitution plasmids that could break and prevent the entry of multiple plasmid types at once (*Adiego-Perez et al., 2019*; *Buckner, Ciusa & Piddock, 2018*; *Kamruzzaman et al., 2017*).

## CONCLUSIONS

CRISPR antimicrobial is a promising approach for combating the rise of antibiotic resistance. To be effective, this approach requires efficient DNA delivery to bacteria population. Compared to phage transduction, plasmid conjugation can deliver DNA to a broader host range but often suffer from low delivery efficiency. We devised multi-plasmid systems for delivering CRISPR antimicrobial, self-transmissible helper vector and replacing plasmid to bacterial population. To the authors' knowledge, our work is the first report on 100% CRISPR delivery efficiency and target plasmid elimination in bacterial population at low donor-to-recipient ratio (1:180) without resorting to antibiotic selection. Unlike any previous work, the target gene in our study was encoded on a mobilizable plasmid that can spread in the population, making it even more difficult to eliminate. Still, we successfully eliminated and replaced the target plasmid, in all observed recipients, with another plasmid from the same incompatibility group. The presence of this same incompatibility group plasmid and CRISPR plasmid would prevent the target plasmid re-entering. An ability to modify genetic content of the entire microbial population, as demonstrated in our work, will be critical for combating antibiotic resistance problems and enabling next generation microbiome engineering technologies.

## ACKNOWLEDGEMENTS

We would like to thank Dr. Wanilada Rungrassamee for editing the manuscript and Dr. Jim Haseloff as well as PLASWIRES (a Synthetic Biology FP7 European research project) for initial inspiration on bacterial conjugation research. We would like to thank Dr. Harris H. Wang, Dr. Ellen Zechner, Dr. Victor de Lorenzo, Dr. Sutthirat Sitthisak and Dr. C. Peter Wolk for providing plasmids or bacterial strains. The authors thank former students who have participated in this project, particularly, Ms. Palisa Mahachai, Ms. Warapon Onnuam, Ms. Tarit Kaewsuk and Mr. Settawut Buadee. The authors thank the Faculty of Medical Science, Naresuan University for providing instruments.

### Funding

This research was supported by a grant MRG6080177 funded by Thai Research Fund (TRF), Thailand, and a grant PRP6205030370 funded by Agricultural Research Development Agency (ARDA), Thailand. The funders had no role in study design, data collection and analysis, decision to publish, or preparation of the manuscript.

## Grant Disclosures

The following grant information was disclosed by the authors:
Thai Research Fund (TRF), Thailand: MRG6080177.
Agricultural Research Development Agency (ARDA), Thailand: PRP6205030370.

## Competing Interests

The authors declare that they have no competing interests.

## Author Contributions

- Panjaporn Wongpayak conceived and designed the experiments, performed the experiments, analyzed the data, prepared figures and/or tables, authored or reviewed drafts of the paper, and approved the final draft.
- Orapan Meesungnoen conceived and designed the experiments, performed the experiments, prepared figures and/or tables, and approved the final draft.
- Somchai Saejang conceived and designed the experiments, performed the experiments, prepared figures and/or tables, and approved the final draft.
- Pakpoom Subsoontorn conceived and designed the experiments, analyzed the data, prepared figures and/or tables, authored or reviewed drafts of the paper, and approved the final draft.

## Data Availability

The raw data are available in the Supplemental Files.
Python code for simulating plasmid transfer dynamics is also available as a Supplemental File.

## Supplemental Information

Supplemental information for this article can be found online at http://dx.doi.org/10.7717/peerj.11996#supplemental-information.

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
