# Peer review of "A highly effective and self-transmissible CRISPR antimicrobial for elimination of target plasmids without antibiotic selection"

_PeerJ, doi:10.7717/peerj.11996_

## Round 0.1 · original submission · Major Revisions

The manuscript has been reviewed by two experts; both found the manuscript interesting and worthy of eventual publication in PeerJ. However, the manuscript will require revision. Reviewer #1 requests testing the plasmid replacement system with further plasmids, and in both directions. I feel including these experiments in the revised manuscript is warranted. Reviewer #2 suggests testing conjugation efficiency under different conditions. While the results of these might be interesting, I do not feel these experiments are necessary to validate the claims of the paper and performing these will be left to the discretion of the authors.
Reviewer #1 also requests thorough statistical analysis of the results. This should be included in the revision. Both reviewers point to a number of small typographical and grammatical errors that should also be corrected. Reviewer #1 further requests a more consistent approach to nomenclature. Please address these and the other minor points raised by the reviewers (see below).

Reviewer 1 ·

Basic reporting

In this paper, Wongpayak et al. demonstrate a CRISPR-based approach to plasmid elimination in target cells, with the aim of combating antimicrobial resistance (AMR). The authors include a sufficient amount of raw data as supplementary information.

The paper is well-structured and well-referenced. Sufficient context for the work is provided in the Introduction, particularly highlighting specific problems with existing literature in the field in a convincing manner, clearly providing a good rationale for this study. The paper is self-contained and the authors write in clear and appropriate English throughout, though there are some instances of incorrect tense (e.g. line 23, “suffer” rather than “suffers”) and missing words (e.g. line 27, “targeted plasmid” should be “the targeted plasmid”) throughout the document. The manuscript would therefore benefit greatly from being proofread by a native-level English speaker.

The data presented are complex and the authors make good use of colour to distinguish strains and cell types in their graphs. However the authors tend to switch between different nomenclature for their plasmids (using what I assume to be a general term, e.g. pCRISPR, and a specific term, e.g. pgNDM1) throughout the manuscript – even from one sentence to the next within a single experiment. This is presumably done in order to reduce jargon, however I found that it actually made things harder to understand. The manuscript would be more readable if a consistent naming approach was maintained throughout (e.g. drop the generalised terms and use the actual plasmid names in all cases).

MINOR ERRORS
General: consider defining the CRISPR acronym upon first mention in the abstract and main text
Line 48: consider a more precise word than “breaks”, e.g. “cleaves”
Line 51: there is some kind of formatting error on this line
Line 56: there should be a comma before Staphylococcus
Line 74: Have the authors considered the presence of pre-existing surface exclusion factors that prevent conjugation?
Line 114: plasmid “pgEMPTY” is given later as “pEMPTY”, presumably this is a typo
Line 150: “1.5 tube” should presumably be “1.5 mL tube”
Line 270: Typo in pCRISPR
Line 273: “incompability” should be “incompatibility”
Throughout: “resistant” should be “resistance” in a few places
Throughout: there should be a space between “E.” and “coli”
Throughout: please use the symbol μ rather than u where appropriate (e.g. μg/mL, μL)
Throughout, including figure legends: genes and DNA sequences should be italicised
Results: I felt that subheadings would improve the readability of this section, if appropriate to the journal style
Figure 2B: there is a colour discrepancy between the key and the graph itself
Figure 4: formatting errors (the figure labels A/C have been cut off)

Experimental design

The design of the conjugation experiments themselves seem appropriate and the authors have included several steps that are fully reproducible from the instructions given; e.g. the number and amount of time that samples were vortexed. This is commendable.

Line 145-146: Donor and recipient cells were adjusted to OD600 ~ 0.3 prior to mixing, but it is unclear whether the cells are actually present at an equal concentration (i.e. CFU/mL). The authors do say in the methods that the 1:100 ratio is “by volume” rather than “by cell number”, which is appropriate, but elsewhere in the manuscript the implication is that cell numbers have been taken into account (e.g. the abstract: “at a low (1:100) donor-to-recipient ratio”). Since different strains have different OD-to-CFU relationships, this confounding factor must be explicitly addressed in order to make the claims about donor-to-recipient ratio valid.

As noted below, there is a complete lack of statistical analysis presented throughout the paper.

Validity of the findings

The major flaw of this study is the clear lack of statistical analysis. Given the nature of the data presented, this needs to be addressed prior to drawing conclusions.

Line 175-179, figure 1c: No statistical analysis is given to confirm the result here, though the data do seem to qualitatively support the claim.

Lines 193-202: Again, statistical analysis is required to confirm these conclusions. The results here (especially the red dots) are difficult to interpret. The claims made by the authors are not supported without thorough statistical analysis, taking into account multiple comparisons.

Lines 213-234: No statistical analysis is given to confirm these results, though the data do seem to qualitatively support the claims.

Lines 219-250: As above… These data are interesting but need statistical analysis to back them up.

Lines 263-4: The authors claim “pSubst by itself is unlikely to be able to enter, compete and replace pTarget that has already established in recipient cells”. However, this is not entirely clear; see below.

Line 278: The authors mention “not significantly different” but again do not present the results or type of statistical test used.

Lines 272-288: The authors try to address the question of whether a new plasmid can establish itself in a population that is already saturated with another plasmid from the same incompatibility group, but this is only tested in one direction with one pair of plasmids. The results are not broadly applicable. Given the tools on hand, the authors could have expanded this experiment to try a few different combinations before reaching conclusions here.

Lines 308-309: As above, the authors claim that “since pTarget and pSubst are from the same incompatibility group, pSubst cannot infect a cell that already has pTarget and vice versa”. To my understanding they haven’t tested it in both directions, so “and vice versa” cannot be claimed.

Figure 4C: I’m not sure what the authors mean by “…and plasmid replacement…” in the title for 4C. Is that evidenced by their experiment/data?

Lines 323-330: Again, no statistical analysis is presented.

Lines 331-332: The authors present a computer simulation in their supplementary material. It seems extensive – I see no reason why this could not be included in the main paper if the results are relevant.

Reviewer 2 ·

Basic reporting

No comments

Experimental design

No comments.

Validity of the findings

No comment.

Additional comments

The manuscript by Wongpayak et al., studies dispersal of conjugative and mobilizable plasmids in E. coli hosts as well as the elimination of genetic target via CRISPR-Cas9 within recipient cells. To this reviewer, the manuscript appears to be of high quality. All the experiments are well-designed and -described. The study demonstrates that with specific plasmid combination a notable portion of the bacterial community can receive the plasmid and hence allow effective manipulation of the community even in the absence of selection for the desired plasmid.
However, conjugation experiments were carried out on nitrocellulose membranes where conjugation is very efficient. It would be nice to see what kind of recipient ratios would be achieved when conjugation is happening in liquid media and with/without shaking. Screening other conditions would also be valuable for the scientific community, like different nutrient ratios (e.g. high and low). These would help define what are the limitations of the observed number of recipients if there are any. Moreover, one practical limitations of the here-presented system is that one should know beforehand exactly the type of plasmid that needs to be replaced. In many instances, there are several plasmids present in e.g. ESBL-bacteria and as such targeting just one type of a plasmid may fail. The authors should discuss these topics in the manuscript. And overall, it should be mentioned that there is a need for in vivo and multi-species studies with these types of mobilizable CRISPR-plasmids in order to find the real limits and the potential of plasmid-based modification of heterogenous bacterial communities. The language in the manuscript is overall good, but there are some small errors that should be fixed. With these changes, this reviewer would support the publication.

---

## Round 0.2 · Minor Revisions

The authors have responded well to the criticisms raised by the reviewers in their revision. However, reviewer #1 notes that the statistical analysis is not always appropriate, especially using t-test for multiple comparisons, where ANOVA with a post-hoc test would be the correct tool. I also agree with the reviewer that colony PCR is not a reliable method to estimate gene copy numbers. Please make the requested alterations and resubmit a revised version.

Reviewer 1 ·

Basic reporting

The general text has been improved and use of plasmid terminology throughout the manuscript has been made much clearer. I have no further concerns regarding the basic reporting.

Experimental design

I am pleased to say that the authors have addressed a major concern of mine: the OD-to-CFU relationship of the strains. This has now been appropriately defined and explained in the manuscript.

Validity of the findings

Generalisation of the system has been appropriately confirmed with an additional set of experiments. The authors’ use of a different plasmid replicon is particularly useful here, so is to be commended… However, I would disagree with their comment on supplementary lines 224-226 that “thinner” colony PCR bands imply lower target copy number. Band intensity from a colony PCR varies according to many factors and is not an appropriate way to measure plasmid copy number, hence no such conclusions can be drawn here. I suggest the authors simply remove this particular inference from the manuscript.

The authors have also added statistical analysis to the paper throughout, but there are some remaining issues:
1. The use of multiple t-tests to compare values within experiments is not ideal; an ANOVA with post-tests for multiple comparisons would be more appropriate, particularly in cases where a two-factor analysis is being performed.
2. There are some figure legends where the statistical indicators (*, **, etc) are not listed for each indicator that appears in the figure. For example, Figure 4 includes NS, ** and *** indicators, but only NS and ** are defined in the legend.
3. A full statistical methodology section could be added to the Materials & Methods.

---

## Round 0.3 · accepted · Accept

The manuscript is now acceptable for publication.